# Protein Networks Associated with Native Metabotropic Glutamate 1 Receptors (mGlu_1_) in the Mouse Cerebellum

**DOI:** 10.3390/cells12091325

**Published:** 2023-05-05

**Authors:** Mahnaz Mansouri, Leopold Kremser, Thanh-Phuong Nguyen, Yu Kasugai, Laura Caberlotto, Martin Gassmann, Bettina Sarg, Herbert Lindner, Bernhard Bettler, Lucia Carboni, Francesco Ferraguti

**Affiliations:** 1Department of Pharmacology, Medical University of Innsbruck, 6020 Innsbruck, Austria; mahnaz.mansouri@gmail.com (M.M.); kasugaiyu@yahoo.co.jp (Y.K.); 2Institute of Medical Biochemistry, Protein Core Facility, Medical University of Innsbruck, 6020 Innsbruck, Austria; leopold.kremser@i-med.ac.at (L.K.); bettina.sarg@i-med.ac.at (B.S.); herbert.lindner@i-med.ac.at (H.L.); 3Dennemeyer, 1274 Hesperange, Luxembourg; nguyentp.dr@gmail.com; 4Centre for Computational and Systems Biology (COSBI), The Microsoft Research University of Trento, 38068 Rovereto, Italy; laura.caberlotto@evotec.com; 5Department of Biomedicine, Pharmazentrum, University of Basel, 4056 Basel, Switzerland; matin.gassmann@unibas.ch (M.G.); bernhard.bettler@unibas.ch (B.B.); 6Department of Pharmacy and Biotechnology, Alma Mater Studiorum University of Bologna, 40126 Bologna, Italy; lucia.carboni4@unibo.it

**Keywords:** glutamate receptors, cerebellum, KCTD12, immunoprecipitation, proteomics

## Abstract

The metabotropic glutamate receptor 1 (mGlu_1_) plays a pivotal role in synaptic transmission and neuronal plasticity. Despite the fact that several interacting proteins involved in the mGlu_1_ subcellular trafficking and intracellular transduction mechanisms have been identified, the protein network associated with this receptor in specific brain areas remains largely unknown. To identify novel mGlu_1_-associated protein complexes in the mouse cerebellum, we used an unbiased tissue-specific proteomic approach, namely co-immunoprecipitation followed by liquid chromatography/tandem mass spectrometry analysis. Many well-known protein complexes as well as novel interactors were identified, including G-proteins, Homer, δ2 glutamate receptor, 14-3-3 proteins, and Na/K-ATPases. A novel putative interactor, KCTD12, was further investigated. Reverse co-immunoprecipitation with anti-KCTD12 antibodies revealed mGlu_1_ in wild-type but not in KCTD12-knock-out homogenates. Freeze-fracture replica immunogold labeling co-localization experiments showed that KCTD12 and mGlu_1_ are present in the same nanodomain in Purkinje cell spines, although at a distance that suggests that this interaction is mediated through interposed proteins. Consistently, mGlu_1_ could not be co-immunoprecipitated with KCTD12 from a recombinant mammalian cell line co-expressing the two proteins. The possibility that this interaction was mediated via GABA_B_ receptors was excluded by showing that mGlu_1_ and KCTD12 still co-immunoprecipitated from GABA_B_ receptor knock-out tissue. In conclusion, this study identifies tissue-specific mGlu_1_-associated protein clusters including KCTD12 at Purkinje cell synapses.

## 1. Introduction

Metabotropic glutamate receptors (mGlus) are members of class C G-protein coupled receptors (GPCRs) activated by glutamate, the major excitatory neurotransmitter in the central nervous system. These receptors are involved in many physiological functions including neuronal excitability, development, synaptic plasticity, and memory [1]. The eight members of this protein family are classified into three groups. Group I consists of mGlu_1_ and mGlu_5_, which share about 70% sequence homology and mainly couple to Gαq. Group I mGlus are selectively activated by (S)-3,5-dihydroxyphenylglycine and are mainly localized post-synaptically [2,3]. In the cerebellar cortex, mGlu_1_ is highly expressed in Purkinje cells and a subset of interneurons, whereas mGlu_5_ is expressed in Golgi and Lugaro cells and in deep cerebellar nuclei [4,5,6,7]. At glutamatergic synapses of Purkinje cells, mGlu_1_ contributes to long-term depression (LTD), which is important for cerebellar learning mechanisms [8]. Gene-targeted deletion of mGlu_1_ results in impaired LTD and severe ataxia [9,10]. This receptor also plays an important role in the elimination of multiple climbing fiber innervation to Purkinje cells during development [11,12]. These functions critically depend on the coupling of mGlu_1_ to Gαq proteins [12,13]. In addition, mGlu_1_ is involved in synaptic plasticity at GABAergic synapses such as rebound potentiation which is mediated by coupling of the receptors to Gαs [14]. A number of studies have shown that G-protein dependent as well as G-protein independent functional properties of mGlu_1_ depend on their interaction with scaffolding and signaling proteins, including other GPCRs and ion-channels [15,16]. Alternative splicing at the mGlu_1_ gene (*Grm1*) generates four variants, namely mGlu_1_α, mGlu_1_β, mGlu_1_γ, and mGlu_1_δ, which share a large part of the N-terminal sequence but differ primarily in their intracellular C-terminal domains [1]. The mGlu_1_α isoform has the longest C-terminal domain and can physically interact with a variety of proteins through motifs that are not present in the shorter isoforms [17,18]. CFTR-associated ligand (aka Golgi-Associated PDZ And Coiled–Coil Motif-Containing Protein) [19], Homer proteins [20,21], Norbin (neurochondrin) [22,23], protein phosphatase 1C [24], Siah-1A [25], and Tamalin [26] are some of the signaling and scaffolding proteins that were reported to directly bind to mGlu_1_α [15]. This isoform was also found to form functional complexes with the Gluδ2 receptor and the short Transient Receptor Potential Cation channel C3 (TRPC3) [27,28,29,30,31].

Most of our current knowledge about mGlu_1_ interaction partners is obtained from affinity purifications or yeast two-hybrid screenings. In recent years, proteomic studies have emerged as a valuable tool for studying the co-assembly of proteins in native tissue. Proteomic approaches have the advantage of identifying stable and transient protein–protein interactions, defining native protein complexes, and finding novel interaction partners [32,33].

In the current study, we used a proteomic approach to identify protein complexes that are associated with mGlu_1_α in the cerebellum. We immunoprecipitated mGlu_1_α from mice cerebellar lysates and analyzed co-purified proteins by mass spectrometry. Using this approach, we identified multiple well-known as well as novel interactors and generated a mGlu_1_ protein interaction network. We investigated a novel mGlu_1_α interaction partner, namely the Potassium Channel Tetramerization Domain-containing protein 12 (KCTD12), a GABA_B_ receptor auxiliary subunit, using in vivo and in vitro methods. Our findings showed that mGlu_1_ and KCTD12 co-exist in the same nanodomain in Purkinje cell spines, though their interaction does not depend on direct physical binding but most likely through interposed proteins.

## 2. Materials and Methods

### 2.1. Experimental Animals

For immunoprecipitations of mGlu_1_α from the cerebellum, adult male and female C57BL/6N wild-type (WT) (n = 23), *Grm1*-knock-out (KO) (n = 7), BALB/c WT (n = 4), GABA_B1_ (n = 4), and GABA_B2_ (n = 4) KO mice were used. C57BL/6N WT (n = 3) and KCTD12-KO (n = 3) adult male and female mice were used to immunoprecipitate KCTD12 from the cerebellum.

The following mice were used: C57BL/6N (Charles River, Sulzfeld, Germany), *Grm1*-KO (gift from GlaxoSmithKline), GABA_B1_ KO [34], GABA_B2_ KO [35], Balb/c littermate, and KCTD12-KO mice [36]. Experimental procedures on animals were approved by the Austrian Animal Experimentation Ethics Board (GZ66.011/28-BrGT/2009) and by the Veterinary Office of Basel-Stadt and were in compliance with the European Convention for the Protection of Vertebrate Animals used for experimental and other scientific purposes, the Animal Experiments Act 2012 (TVG 2012), and the EU Directive 2010/63/EU. The authors further attest that all efforts were made to minimize the number of animals used.

### 2.2. Antibodies and Vectors

Antibodies against mGlu_1_α (Af811-1 and Af660-1) were purchased from Frontier Institute Co. Ltd. (Hokkaido, Japan) and KCTD12 antibodies were generated as previously reported [37].

The plasmid containing the open reading frame for the mouse mGlu_1_α DNA and lentiviral particles packaged for mouse KCTD12 DNA was obtained from Genecopoeia (Rockville, MD, USA). The expression of KCTD12 was under the control of the cytomegalovirus (CMV) promoter and both the plasmid and lentiviral vectors contained the neomycin resistance gene. The plasmid containing the open reading frame for mouse GABA_B2_-YFP DNA under the control of the CMV promoter was previously reported [38].

### 2.3. Immunoprecipitation of Proteins from Mouse Cerebellum (P2 Fraction)

To immunoprecipitate mGlu_1_α and KCTD12 and mouse cerebella were dissected and pooled. Homogenization was performed in ice-cold 10 mM Tris-HCl, pH 7.4 buffer containing 320 mM sucrose, 1 mM phenylmethylsulphonyl fluoride, 1 mM NaF, 1 mM Na_3_VO_4_, and complete EDTA-free protease inhibitors (Roche, Vienna, Austria) using a motorized homogenizer (Sartorius, Göttingen, Germany) at a speed of 1400 rpm and 10 strokes. Lysates were centrifuged for 10 min at 1000× *g* and 4 °C to remove unbroken cells and tissue debris. This process was repeated 2 more times. Supernatants were collected and centrifuged for 40 min at 17,000× *g* at 4 °C. The pellet (P2 fraction) was resuspended in ice-cold 25 mM Tris-HCl, pH 7.4 buffer containing 1 mM NaF, 1 mM Na_3_VO_4_, and complete EDTA-free protease inhibitors. The protein concentration was determined by the Bradford protein assay using bovine serum albumin as the standard protein.

P2 samples were centrifuged for 30 min at 20,000× *g* and 4 °C. The pellet was suspended for 1 h in ice-cold 25 mM Tris-HCl, pH 7.4 buffer containing 0.5% sodium deoxycholate, 1% NP-40, 0.1% sodium dodecyl sulfate (SDS), 137 mM NaCl, 3 mM KCl, 1 mM phenylmethylsulfonyl fluoride, 1 mM NaF, 1 mM Na_3_VO_4_, and complete EDTA-free protease inhibitors. Detergent lysates were then centrifuged for 60 min at 20,000× *g* and 4 °C. The supernatant was aspirated carefully and incubated with primary antibodies (0.2 µg/100 µg proteins) for 2 h at 6 °C with constant rotation. Dynabeads Protein-A (Invitrogen, Waltham, MA, USA) were added at the ratio of 1 µL:20 µg of protein followed by incubation for 1 h at 6 °C. The immunoprecipitation eluates were obtained by heating the samples in 1× Laemmli buffer plus 20 mM dithiothreitol for 10 min at 70 °C and 5 min at 56 °C to prevent oligomerization.

### 2.4. Immunoblotting

Samples were loaded on NuPAGE Bis-Tris 4–12% precast gels and proteins were resolved at 80 V in MOPS SDS buffer. Proteins were transferred to a polyvinylidene difluoride membrane overnight at 150 mA, 6 °C. The membranes were stained with Ponceau-S for 10 min and incubated in 5% dry milk blocking solution for 1 h at room temperature. Membranes were then incubated in primary antibodies for 2–3 overnights and 6 °C (1:3000). Immunoreactive bands were detected by incubation in horseradish peroxidase-conjugated secondary antibodies (Invitrogen) followed by the ECL Prime reagent. Chemiluminescence was visualized with the Fusion SL-4 Vilber Lourmat imaging system (Peqlab, Erlangen, Germany).

### 2.5. Mass Spectrometry (LC–MS/MS) Analysis of Proteins

Eluates were lyophilized in a Christ RVC 2-18 concentrator and loaded on precast NuPAGE 4–12% Bis-Tris gels (Invitrogen). After 1.5 cm running in a MOPS SDS buffer, the gels were stained for 1 h in Coomassie R-250 at room temperature. Destaining was performed overnight in 40% methanol and 10% glacial acetic acid solution.

Protein bands were excised from gels and digested with trypsin obtained from porcine pancreas (Sigma-Aldrich, Vienna, Austria), as previously described [39]. Tryptic digests were analyzed using an UltiMate 3000 nano-HPLC system coupled to an LTQ Orbitrap XL mass spectrometer (Thermo Scientific, Bremen, Germany). The peptides were separated on a homemade fritless fused silica micro-capillary column (75 µm i.d. × 280 µm o.d. × 10 cm length) packed with 3 µm reversed phase C18 material (Reprosil). The solvent for HPLC was 0.1% formic acid (solvent A) and 0.1% formic acid in 85% acetonitrile (solvent B). The gradient profile was as follows: 0–2 min, 4% B; 2–55 min, 4–50% B; 55–60 min, 50–100% B; and 60–65 min, 100% B. The flow rate was 250 nL/min.

The LTQ Orbitrap XL mass spectrometer was operated in the data-dependent mode selecting the top 10 most abundant isotope patterns with a charge of 2+ or 3+ from the survey scan with an isolation window of 2 for the mass-to-charge ratio (*m*/*z*). Survey full scan MS spectra were acquired from 300 to 2000 *m*/*z* at a resolution of 60,000 with a maximum Injection Time (IT) of 20 ms and automatic gain control (AGC) target 1 × 10^6^. The selected isotope patterns were fragmented by collisional-induced dissociation (CID) with a normalized collision energy of 35 and a maximum injection time of 55 ms.

Data analysis was performed using Proteome Discoverer 1.3 (Thermo Scientific) with the search engine Sequest. The raw files were searched against the *Mus musculus* database (167,940 entries) extracted from the NCBInr. The precursor and fragment mass tolerances were set to 10 ppm and 0.02 Da, respectively, and up to two missed cleavages were allowed. Carbamidomethylation of cysteine and oxidation of methionine were set as variable modifications. Peptide identifications were filtered at a 1% false discovery rate.

### 2.6. Generation of a Stable Cell Line Expressing KCTD12

HEK293 cells were seeded for 24 h and transduced with lentiviral particles containing KCTD12 DNA at a multiplicity of infection-10 and selected with G418 (600 μg/mL). After 48 h, the cells were trypsinized and transferred to a new dish. G418 was added to the medium for the next 3 passages until dead cells were sparse. From that stage onwards, cells were split regularly when almost 90% were confluent without adding G418 to the growth medium. Cells were maintained in an incubator at 37 °C, 5% CO_2_, and 95% humidity. Confirmation of KCTD12 protein translation was obtained by Western blot and immunofluorescence.

### 2.7. Immunofluorescence

KCTD12-expressing cells were quickly washed with ice-cold PBS and fixed in ice-cold 4% paraformaldehyde for 45 min. Fixed cells were washed 3 times in tris-buffered saline followed by 1 h incubation in blocking solution at room temperature. Cells were incubated overnight with primary antibodies (rabbit anti-KCTD12 1:3000; guinea pig anti mGlu_1_α 1:1000) at 6 °C. Cells were washed three times with tris-buffered saline and incubated with secondary antibodies [Cy3 donkey-anti rabbit IgG 1:400 (Jackson ImmunoResearch, Ely, UK); Alexa 488-goat anti guinea pig IgG, 1:1000 (Invitrogen)] overnight at 6 °C in the dark. Cells were washed twice in tris-buffered saline, a coverslip was mounted using Vectashield, and analysis was performed using a Zeiss Axioimager M1 microscope equipped with a metal halide lamp.

### 2.8. Transient Transfection and Immunoprecipitation

Stable cell lines were transfected with plasmids containing the mGlu_1_α DNA and the GABA_B2_-YFP DNA. DNA and lipofectamine (Invitrogen) were diluted in Optimem, mixed at a ratio of 1:3, and incubated for 20 min at room temperature. The mixtures were added to the growth medium and the cells were kept in an incubator for 72 h. The cells were washed with warm PBS and lysed in an ice-cold buffer used for immunoprecipitation. Lysates were centrifuged for 40 min at 17,000× *g* and 4 °C. The supernatant was aspirated carefully and KCTD12 was immunoprecipitated using a guinea pig polyclonal antibody raised against KCTD12 following the same protocol as described above.

### 2.9. Freeze-Fracture Replica Immunogold Labeling (FRIL)

FRIL was performed according to previously published procedures [40,41]. Two C57BL/6N WT and two KCTD12-KO mice were perfused transcardially for 10 min with a solution containing 1% paraformaldehyde and 15% of a saturated solution of picric acid in 0.1 M phosphate buffer at a rate of 5 mL/min. The brains were quickly extracted from the skull and the cerebellums were cut with a vibratome (Leica Microsystems VT1000S, Vienna, Austria) into 140 µm thick coronal sections from where samples of the cerebellar cortex molecular layer (Crus 2 lobule) were dissected out under a stereomicroscope. The slices were cryoprotected in 30% glycerol in 0.1 M phosphate buffer and high-pressure frozen by means of an HPM 010 machine (Bal-Tec, Balzers, Liechtenstein). The frozen slices were then freeze-fractured at −115 °C and replicated with a first layer of carbon (5 nm), shadowed by platinum (2 nm), and followed by a second carbon layer (15 nm) in a freeze-etching BAF 060 device (Bal-Tec). After thawing, the tissue attached to replicas was digested with stirring at 80 °C overnight in SDS solubilization buffer for FRIL.

For immunogold labeling of replicas, blocking was performed first in a solution consisting of 5% bovine serum albumin and 0.1% Tween-20 in tris-buffered saline, pH 7.4 at room temperature. Replicas were then incubated in primary antibodies (guinea pig anti-KCTD12 1:90; rabbit anti mGlu_1_α 1:300) for 72 h at 6 °C. Following extensive washes and 1 h blocking, replicas were incubated with gold-conjugated secondary antibodies of 5 and 10 nm (British Biocell International, Cardiff, UK; BBI 1175; BBI 009076) for 48 h at 6 °C. Incubation of the primary and secondary antibodies was in a sequence with the antibodies for the detection of mGlu_1_α always incubated first. The specificity of the antibodies was tested on tissue obtained from KO mice as well as by omitting the primary antibody. Replicas were mounted on pioloform-coated mesh copper grids and examined with a Philips CM120 transmission electron microscope (Philips, Eindhoven, The Netherlands).

### 2.10. Network Construction

We constructed a Protein–Protein Interaction (PPI) network for immunoprecipitated proteins identified in all 6 replicate experiments. To obtain reliable protein interactions, we extracted only the validated interactions for mice. The mouse network was modelled using the 1-step neighbors of the putative interacting proteins with high confidence. We comprehensively merged the interactions from multiple interaction databases, including iRefIndex (https://irefindex.vib.be; accessed on 12 September 2022), Mentha (https://mentha.uniroma2.it; accessed on 12 September 2022), InnateDB-all (https://www.innatedb.com; accessed on 19 September 2022), EBI-GOA (https://www.ebi.ac.uk/GOA/index; accessed on 19 September 2022), MiNT (https://mint.bio.uniroma2.it; accessed on 12 September 2022), IMEX (https://www.imexconsortium.org; accessed on 19 September 2022), and IntAct (https://www.ebi.ac.uk/intact/home; accessed on 19 September 2022). Indeed, the seven interaction databases cover the mouse interactome. To ensure data quality, the constructed network consists only of non-redundant and validated protein interactions. To this aim, we extracted protein interactions from each database and then combined them into the final network after removing overlapping proteins. The network was visualized with the Cytoscape software 3.9.1 (https://cytoscape.org accessed on 19 September 2022).

### 2.11. Network Analysis

There is evidence that the functional importance of proteins might be inferred from their topological properties, specifically their key positions in the protein interaction network [42,43]. To gain information on the network and its participating proteins, two centrality indices were evaluated for each protein: the degree and betweenness. The degree centrality shows how many direct neighbors a node in the network connects to. A protein is considered as a hub in the network if its degree centrality is high. Betweenness shows the bridge role of a protein for other proteins in the network [44]. A protein presenting high betweenness centrality is an important connector in the network, playing a key mediator role. The degree centrality is measured by counting the neighborhood of a node in the network, thus identifying network hubs. Betweenness is the centrality measure based on shortest-path calculations.

A given network *G*(*N*,*E*) consists of a set of nodes (*N*) and a set of edges (*E*) between them. An edge *e_ij_* connects node *n_i_* with node *n_j_*. In this paper, the extracted network is unweighted and undirected. In an undirected graph, *e_ij_* and *e_ji_* are considered identical. Therefore, the neighborhood ℵ*_i_* for a node *n_i_* is defined as its direct connected neighbors, as follows:(1)$$ℵ=\left\{n_{j}\;:\;e_{ij\;}\in E\right\}$$

The degree *Di* of a node is defined as the number of nodes |ℵ*_i_*| in its neighborhood ℵ*_i_*.

The betweenness centrality is calculated based on the shortest paths. For each node *n_k_* in the network, we counted the total number of shortest paths from node *n_i_* to node *n_j_*, called *p(v_i_*, *v_j_)* as well as the number of those paths that pass through *n_k_*, called *p(v_i_*, *v_j_*, *v_k_)*. The betweenness *B(v_k_)* of a node *v_k_* is defined as follows:(2)$$B\left(n_{k}\right)=\underset{n_{i}\neq n_{k}\neq n_{j}}{\sum }\frac{p\left(n_{i},\;n_{k},\;n_{j}\right)}{d\left(n_{i},n_{j}\right)}$$

Pathway analysis and the identification of enriched GO terms were performed using DAVID and GOrilla based on the 10% highest scoring proteins in the network [45,46,47].

## 3. Results

### 3.1. Immunoprecipitation of mGlu1α from Mouse Cerebellum

To identify protein complexes that can associate with mGlu_1_α, we immunoprecipitated the receptor from mouse cerebellar extracts (Figure 1A). Pilot experiments were performed to determine the optimal conditions for immunoprecipitation. Two main conditions have been optimized: (1) solubilization and (2) the choice and concentration of antibody. Optimal detergent conditions were determined empirically by assessing the amount of native mGlu_1_α extracted from cerebellar lysates by Western blot. Under our experimental conditions, a combination of 0.5% deoxycholate, 0.1% SDS, and 1% NP-40 was most effective in extracting mGlu_1_α. Three different antibodies were tested for immunoprecipitation: a guinea pig polyclonal antibody raised against the amino acids 945-1127 (Af660-1) within the C-terminal region of mGlu_1_α and two rabbit polyclonal antibodies raised against amino acid residues 945-1127 (Af811-1) or 1179-1199 (AB1551). Only the antibodies directed against the sequence 945-1127 were able to specifically immunoprecipitate mGlu_1_ and were used in subsequent experiments. A concentration of 0.2 µg of antibody per 100 µg of proteins resulted in the most efficient immunoprecipitation of the receptor. Three immunoprecipitation replicate experiments were then performed with antibody Af811-1 and three additional ones with antibody Af660-1. Immunoprecipitations from *Grm1*-KO cerebellar lysates were carried out as controls for both antibodies.

In the LC–MS/MS analysis, mGlu_1_α was successfully identified in all six immunoprecipitation experiments from WT, but not from *Grm1*-KO animals. In all experiments, the highest SEQUEST score was indeed observed for mGlu_1_α. The protein sequence coverage of the receptor was between 31.6 and 49.5% (Figure 1B,C; Table 1). The number of unique peptides specifically corresponding to mGlu_1_α ranged from 7 to 11. Additional 19 to 28 peptides were in common between mGlu_1_α and mGlu_1_β (Table 1; Appendix A). mGlu_1_β is a short alternatively spliced variant of the *Grm1* gene in which the last 318 amino acid residues of the mGlu_1_α variant are replaced by 20 different residues [1]. Available evidence suggests that mGlu_1_α and mGlu_1_β variants heterodimerize both in vitro and in vivo [48,49,50] which affects receptor trafficking to the plasma membrane of Purkinje cell [46,47]. In line with these previous findings, we were able to identify specific mGlu_1_β peptides in five out of six experiments, despite the fact that the specific portion of the mGlu_1_β variant is quite short. Therefore, our results provide further support to the notion that mGlu_1_α and mGlu_1_β heterodimerize in Purkinje neurons.

### 3.2. mGlu1α Co-Immunoprecipitated Proteins

Proteomic analysis using the *Mus musculus* NCBInr database with a 99% confidence threshold yielded hundreds of proteins from each experiment. Immunoglobulins, keratins, and proteins co-immunoprecipitated from lysates of *Grm1*-KO mice were considered as non-specific interacting proteins and, therefore, removed from the list of putative mGlu_1_ interaction partners. We identified a total of 304 proteins in the data pooled from the six experiments (Appendix A).

Among the identified proteins, 25 proteins were consistently detected in all 6 replicate experiments (Table 2) and were, therefore, considered putative interacting proteins with a high confidence. Several of these proteins have been previously described as mGlu_1_ interactor partners, such as Homer proteins, the G-protein α/o subunit, inositol 3-phosphate receptors, and Gluδ2, as well as protein kinase C and calcium/calmodulin-dependent protein kinase II (CaMKII) [1,20,51,52,53]. Other previously recognized mGlu_1_ interactors were detected in this study, although they were not included in Table 2 because they were not identified in all six replicates. Among them, the G-protein α/q subunit, the predominant G-protein coupled to mGlu_1_ [1], was positively detected in four experiments out of six (Appendix A). Likewise, TRPC3 was also identified in four experiments (Appendix A) in agreement with previous reports establishing TRPC3 as a downstream effector of mGlu_1_-dependent synaptic transmission in Purkinje cells [28]. Ca_V_2.1 calcium channels were reported to interact with mGlu_1_ in Purkinje neurons [54] and were among the co-immunoprecipitated proteins (Appendix A).

We further detected mGlu_5_, though only in one experiment, that was shown to form functional heterodimers with mGlu_1_ [55,56] despite the fact that their colocalization is restricted to Golgi and Lugaro cells and deep nuclei in the adult rodent cerebellum [4,5,6,7]. GABA_B_ and GABA_A_ receptor subunits were also recovered (Appendix A), consistent with the identification of mGlu_1_ also at GABAergic synapses [57,58].

Among the proteins identified with high confidence, we also detected novel putative interactors (Table 2). These included the ρ guanine nucleotide exchange factor 33 (RhoGEF33), excitatory amino acid transporters, 14-3-3 adapter proteins, Na^+^/K^+^ ATPase subunits, synaptotagmin-2 (Syt2), the K^+^/Cl^-^ cotransporter KCC2, and the KCTD12 protein. The KCTD family of proteins includes 21 members that share a common tetramerization T1 domain [59]. KCTD8, 12, 12b, and 16 form homo- and hetero-oligomers and directly bind to the GABA_B2_ receptor subunit via their T1 domains [38,60,61]. The expression of KCTD12 and 12b strongly desensitizes the GABA_B_ receptor response [62].

### 3.3. mGlu_1_ Interactome

To gain further insight into the interactions between the identified and other neuronal proteins, we constructed a network based on validated interactions as available from the literature and collected in several databases (see Section 2.10). We constructed a network from the most reliably identified proteins (Table 2). The network had 643 nodes and 964 edges; the section centered on mGlu_1_ is displayed in Figure 2. The highlighted proteins belong to the starting list of high-confidence interactors; they are endowed with high scores in the degree and/or betweenness centralities (Appendix A) to signify close interactions and important mediating roles. Out of the 304 proteins identified by LC–MS/MS analysis, in at least one of the experiments, 26.6% were present among the nodes of the network. Pathway and gene ontology (GO) enrichment analysis was carried out on network proteins (Appendix A). Pathway enrichment analysis, as expected, identified among the terms with highest score: glutamatergic synapse, protein–protein interaction at synapses and long term potentiation. Noteworthy, also circadian entrainment, the cGMP-PKG signaling pathway and neurexins and neuroligins showed high significance. GO term enrichment highlighted the known role of mGlu_1_ in synaptic plasticity, glutamatergic signaling and locomotor behavior [63], but also regulation of Ca^2+^/Na^+^ antiporter activity and mitogen-activated protein kinases [64].

### 3.4. Interaction between mGlu_1_α and KCTD12 in the Cerebellum

We then focused on the putative interaction between mGlu_1_α and KCTD12 because (1) KCTD12 directly interacts with the GABA_B2_ subunit of GABA_B_ receptors [38,62], which are also members of the class C GPCRs and share structural similarities with mGlu_1_ [17]; (2) KCTD12 is highly expressed in the dendritic spines and shafts of Purkinje cells [37,62]; and (3) KCTD12 co-purifies G-protein βγ subunits even in the absence of GABA_B2_ [65]. Interestingly, in our proteomic approach, G-protein βγ subunits were also co-immunoprecipitated with mGlu_1_α.

At first, we performed co-immunoprecipitation experiments followed by Western blots using cerebellar lysates from WT, *Grm1*-KO, and KCTD12-KO mice. KCTD12 was consistently co-immunoprecipitated with mGlu_1_α in WT but not *Grm1*-KO eluates (Figure 3). The co-assembly between the two proteins was further examined by reverse co-immunoprecipitation from cerebellar lysates of WT and KCTD12-KO mice. When KCTD12 was immunoprecipitated, bands corresponding to mGlu_1_α (monomer around 145 kDa and dimer/multimer above 205 kDa) were also detected in WT but not KCTD12-KO eluates (Figure 3). However, the efficiency of mGlu_1_α co-immunoprecipitation with KCTD12 appeared very low and might be due to the preferential and strong binding of KCTD12 to GABA_B_ receptors. Taken together, these results support a direct or indirect interaction of mGlu_1_α with KCTD12 in the cerebellum.

### 3.5. The Co-Existence of mGlu1α and KCTD12 in the Same Microdomain of Cerebellar Purkinje Cell Dendritic Spines

Next, we studied the spatial relationship between mGlu_1_α and KCTD12 in the mouse cerebellar cortex using the freeze-fracture replica immunogold labeling (FRIL) technique. Protoplasmic face immunogold labelling for both mGlu_1_α and KCTD12 was observed at postsynaptic elements of Purkinje cells. At Purkinje cell spines, immunogold particles corresponding to KCTD12 were mainly found peri-synaptically (Figure 4A,B), similar to mGlu1α [58,66]. At dendritic shafts, KCTD12 was also detected at extra-synaptic sites with or without mGlu1α in close vicinity (Figure 4C). The nearest neighbor analysis confirmed that labeling for mGlu_1_α and KCTD12 co-existed in a nano-domain in both dendritic spines and shafts of Purkinje cells (mean distance ± s.e.m.: 26.70 ± 2.0 nm for spines; 36.86 ± 3.44 nm for dendrites) (Figure 4D). The distance between gold particles detecting mGlu_1_α was also analyzed (mean distance ± s.e.m.: 19.73 ± 0.85 nm for spines; 22.90 ± 0.95 nm for dendrites), thus revealing that the distribution of their nearest neighbor distance was shorter when compared to gold particles detecting KCTD12 (two-sample Kolmogorov-Smirnov test, spines: *p* = 0.0028; dendrites: *p* = 0.0001) and that this is probably due to homodimerization (Figure 4D). The specificity of immunogold labeling was tested by labeling replicas obtained from mGlu_1_-KO and KCTD12 KO cerebellum as well as omitting primary antibodies.

### 3.6. In Vitro Analysis of the mGlu1α–KCTD12 Interaction

To further examine the interaction between mGlu_1_α and KCTD12, we adopted an in vitro approach. HEK293 cells stably over-expressing mouse KCTD12 were transiently transfected with a mouse mGlu_1_α plasmid (Appendix A). Since KCTD12 directly binds to the GABA_B2_ receptor subunit [62], cells were transfected with GABA_B2_ as a positive control. Mock-transfected cells were used as a negative control. KCTD12 was successfully immunoprecipitated from whole cell lysates of mock-transfected, mGlu_1_α-transfected, and GABA_B2_-transfected cells (Figure 5). However, there was no co-immunoprecipitation of mGlu_1_α with KCTD12, whereas the GABA_B2_ receptor was detected in the co-immunoprecipitation (Figure 5). These findings suggest that the interaction between mGlu_1_α and KCTD12 is not occurring via a direct binding but is mediated indirectly through additional proteins.

### 3.7. The mGlu1α Receptor–KCTD12 Interaction Is Not Mediated by GABAB Receptors

Since Purkinje cells abundantly express GABA_B_ receptors [67], we investigated whether the in vivo interaction between mGlu_1_α and KCTD12 occurs through binding of KCTD12 to GABA_B_ receptors. We immunoprecipitated mGlu_1_α from cerebellar lysates of WT, GABA_B1_-KO, and GABA_B2_-KO mice. If the interaction between mGlu_1_α and KCTD12 were due to GABA_B_ receptors, then KCTD12 should not be detected in the immunoprecipitants of mGlu_1_α from GABA_B1_-KO and GABA_B2_-KO lysates. However, a band corresponding to the molecular weight of KCTD12 was detected in mGlu_1_α immunoprecipitants from both GABA_B1_-KO and GABA_B2_-KO lysates (Figure 6), suggesting that the interaction of mGlu_1_α with KCTD12 is not mediated by GABA_B_ receptors.

## 4. Discussion

In this study, we determined that the mGlu_1_ interactome at mouse cerebellar synapses comprises 304 proteins forming a dense network involved in a number of neuronal functions including receptor trafficking, intracellular signaling, and synaptic plasticity. Using a highly conservative approach to select a small number of high confidence interacting partners, i.e., detected in all 6 experimental replicates, we could identify 25 proteins that interact directly or indirectly with mGlu_1_, some of which were already known. Among the novel interactors, we focused on KCTD12 because of its direct binding to GABA_B2_ [62] which has strong structural similarities to mGlus [17]. Taking a high-resolution imaging approach, we showed, by means of immuno-electron microscopy, that KCTD12 and mGlu_1_α were present in the same nanodomain in Purkinje cell spines. However, using a recombinant mammalian cell line co-expressing both proteins, we were unable to co-immunoprecipitate mGlu_1_α with KCTD12. We excluded the possibility that the GABA_B_ receptor mediates the interaction between KCTD12 and mGlu_1_α. Taken together, our findings suggest that KCTD12 interacts with mGlu_1_α indirectly, possibly via G-protein subunits [65] or other proteins that interact with mGlu_1_α.

We adopted a proteomic approach based on co-immunoprecipitation and LC–MS/MS. The isolation of protein complexes from brain tissues using immunoprecipitation coupled to MS provides a significant advantage over other methods to detect protein–protein interactions as it allows for the analysis of complexes within native physiological cellular domains. However, proteomic results can be confounded by false positive interactions due to antibody cross-reactivity. To overcome this problem, we have used tissue obtained from *Grm1*-KO mice as control samples. This strategy helped us to ensure the selection of truly interacting proteins expressed in the mouse cerebellum.

Using only interacting partners with high confidence, we built a protein interaction network that had over 600 nodes. However, we are aware that many bona fide interacting proteins might have been excluded from such a conservative approach. Our findings show that approximately one-fourth of the proteins identified by our proteomic approach overlapped with the network proteins, on one hand supporting the validity of the network for cerebellar synapses also. On the other hand, our data indicate a molecular complexity of the cerebellar mGlu_1_ interactome greater than the models generated based on the currently available databases and the extant literature. Future studies can avail the additional identified proteins to expand our knowledge on the mGlu_1_ interactome both in terms of its fundamental as well as synapse-specific molecular components.

Our proteomic approach identified several proteins already described as mGlu_1_ interactor partners, such as Homer proteins that directly interact with the C-terminal domain of group I mGlus [20,68]. Likewise, a number of effectors involved in the classical signaling pathway of group I mGlus, namely the hydrolysis of phosphoinositides and release of Ca^2+^ from intracellular stores, were detected by our LC–MS/MS analysis of co-immunoprecipitated proteins. These effectors included the inositol 3-phosphate receptor type 1, phospholipase Cβ4, calmodulin, protein kinase Cγ, and TRPC3 besides the G-protein αq and αi subunits. Amid the mGlu_1_ high confidence interacting proteins, we also detected subunits of CaMKII, a highly abundant serine/threonine kinase in the post-synaptic density of Purkinje cells. This finding is consistent with studies showing the presence of multiple CaMKII consensus motifs in the mGlu_1_ C-terminal domain [69,70,71] and their phosphorylation by CaMKII through a direct interaction [70]. As for the interactors, we identified members of the 14-3-3 protein family, small-size proteins (27–32 KDa) highly enriched in the brain and involved in multiple cellular signaling events by functioning as adaptors and scaffolds. The 14-3-3 proteins have been reported as important modulators of glutamatergic synapses, potentially integrating multiple signaling pathways [72]. They interact with several G-protein coupled receptors, including GABA_B_ and α2-adrenergic receptors [73] but interactions with mGlu_1_ have never been reported thus far. Nevertheless, a 14-3-3 isoform was shown to mediate the group I mGlu-agonist-induced down-regulation of KCNK3 potassium channel activity through protein kinase C [74], thus supporting the hypothesis of a role in mGlu_1_ signaling. Although with low efficiency (in only one experiment, but the one yielding the highest number of interactors), we detected mGlu_5_ among the interaction partners of mGlu_1_. In the adult rodent cerebellum, only Golgi and Lugaro cells and neurons in the deep nuclei were shown to co-express mGlu_1_ and mGlu_5_ [4,5,6,7]. Group I mGlus are known to form heterodimers in recombinant systems and cultured neurons [55,56] and to co-purify from brain membranes [55,75]. While the existence of functional mGlu_1_/mGlu_5_ heterodimers in native tissue remains to be unambiguously demonstrated, our findings provide further support to this notion. The low efficiency may be compatible with the restricted co-expression of these two receptors in just a few cerebellar cell types.

In our experiments, we consistently detected a high number of unique peptides for the neuron-specific α3 catalytic subunit as well as other subunits of the Na^+^/K^+^-ATPase, which contributes to maintaining the polarity of mammalian cells by actively exporting Na^+^ and importing K^+^ [76]. Despite the fact that the Na^+^/K^+^-ATPase was previously shown to interact with glutamate transporters and ionotropic receptors [76], possible non-specific co-immunoprecipitation cannot be ruled out because of its very high expression on the neuronal plasma membranes. Among the newly discovered interactors, RhoGEF33, which belongs to a membrane-bound protein family of upstream regulators of Rho GTPases, is involved in different cellular processes including GPCR downstream signaling. The role of RhoGEF33 in relation to mGlu_1_ may depend on its Rho GTPase function [77] involved in the regulation of synaptic structure and efficacy [78].

Among the novel putative interactors identified in our study, we focused on KCTD12 because of its direct binding to the GABA_B2_ receptor subunit [62] which has structural similarities to mGlus [17]. We reasoned that KCTD12 could similarly interact with mGlu_1_ and regulate its signaling. Indeed, we were able to co-immunoprecipitate KCTD12 using antibodies against mGlu_1_α. Reverse co-immunoprecipitation experiments similarly allowed the detection of mGlu_1_α, although with lower efficiency. In addition, we showed that mGlu_1_α and KCTD12 colocalize in the same nanodomain in Purkinje cell spines, strengthening the idea of a functional interaction in neurons in vivo. However, our nearest neighbor analysis suggested that the distance between KCTD12 and mGlu_1_α was incompatible with a direct physical interaction. Consistent with this hypothesis, we were unable to co-immunoprecipitate mGlu_1_α with KCTD12 from HEK293 cell extracts expressing these two proteins. Previous studies reported a functional interaction between mGlu_1_α and GABA_B_ receptors at Purkinje cell synapses [79,80] and showed that the activation of GABA_B_ receptors facilitates the mGlu_1_-mediated LTD induction [51,80,81]. Furthermore, the two receptors were shown to co-immunoprecipitate from cerebellar extracts [51,79]. However, their interaction was shown to be indirect and mediated via Gβγ subunits released upon GABA_B_ receptor activation [82]. In line with these findings, we observed that KCTD12 was co-immunoprecipitated with the mGlu_1_α even in the absence of the GABA_B1_ and GABA_B2_ receptor subunits, suggesting that the interaction between KCTD12 and mGlu_1_α is independent of GABA_B_ receptors. A likely candidate to mediate the interaction between mGlu_1_α and KCTD12 is the Gβ subunit of the G protein. KCTD12 also binds to Gβ [38] independently of GABA_B_ receptors [65]. It is, therefore, conceivable that mGlu_1_ binds G proteins bound to KCTD12. This would also explain the weak co-immunoprecipitation of mGlu_1_ with anti-KCTD12 antibodies. Future studies should explore whether Gβ subunits do indeed mediate this interaction.

Irrespective of the interaction mechanism between mGlu_1_α and KCTD12, a still open question is the potential functional role of this interaction. KCTD12 mediates fast GABA_B_ receptor desensitization through its binding to the activated Gβγ subunits [65], thereby uncoupling them from the inwardly rectifying K^+^ (GIRK or Kir3) and voltage-gated Ca^2+^ channels (VGCCs) modulated by GABA_B_ receptor activation [62]. It can be surmised that KCTD12 similarly modulates mGlu_1_ complex postsynaptic responses. In Purkinje cells, both VGCCs, namely Ca_V_2.1 and Ca_V_3.1 [54,83], and potassium channels [84,85] were shown to functionally interact with mGlu_1_. KCTD12 may thus regulate mGlu_1_α-mediated responses through these channels. In this respect, it is interesting to note that the related KCTD8 and KCTD12b proteins were shown to directly bind to and co-localize with Cav2.3 channels at the presynaptic zone in projections from the medial habenula where they modulate neurotransmitter release [86]. Future studies will be necessary to clarify the functional significance of the mGlu_1_α–KCTD12 interaction.

## 5. Conclusions

The mGlu_1_ interactome presented here serves as a molecular framework for further exploring the signaling, trafficking, and involvement in the cerebellar function of this receptor. Some of the proteins identified by our proteomic approach are novel putative interactors whereas others were already known to interact directly or indirectly with mGlu_1_. Among the novel interactors, we focused on KCTD12 and showed that it coexists with mGlu_1_α in the same nanodomain in Purkinje cell spines. However, biochemical analyses suggested an indirect interaction via additional proteins that warrants follow-up functional investigations.

## Figures and Tables

**Figure 1 cells-12-01325-f001:**
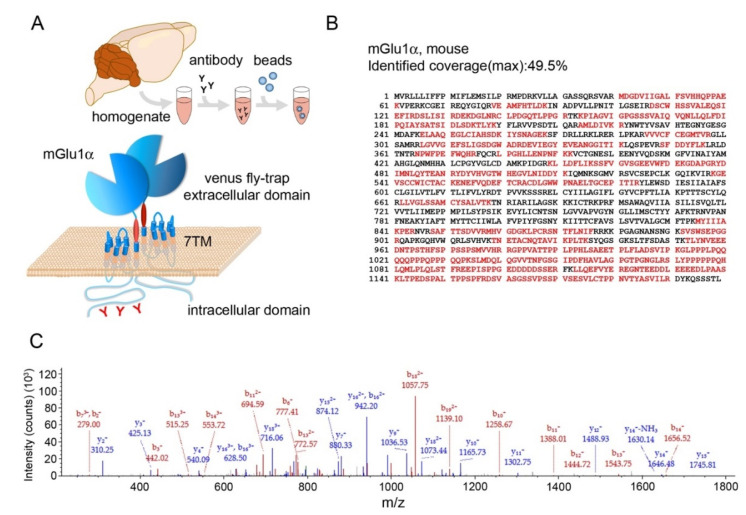
Multiepitope affinity purification identifies native mGlu_1_ in mouse cerebellar protein extracts. (**A**) Schematic of the immunoprecipitation procedure and of the mGlu_1_α structure. (**B**) Primary mGlu_1_α amino acid sequence. Red indicates the protein coverage of mGlu_1_α identified by MS analysis. (**C**) MS/MS spectra of the mGlu_1_ unique peptide YDYVHVGTWHEGVLNIDDYK. Monoisotopic m/z 808.38007 Da (−1.56 mmu/−1.93 ppm). MH+: 2423.12564 Da, RT: 32.51 min. Fragment match tolerance used for search: 0.8 Da. Fragments used for search: b; b-H_2_O; b-NH_3_; y; y-H_2_O; and y-NH_3_. Ion series: b (shown in red), y (shown in blue).

**Figure 2 cells-12-01325-f002:**
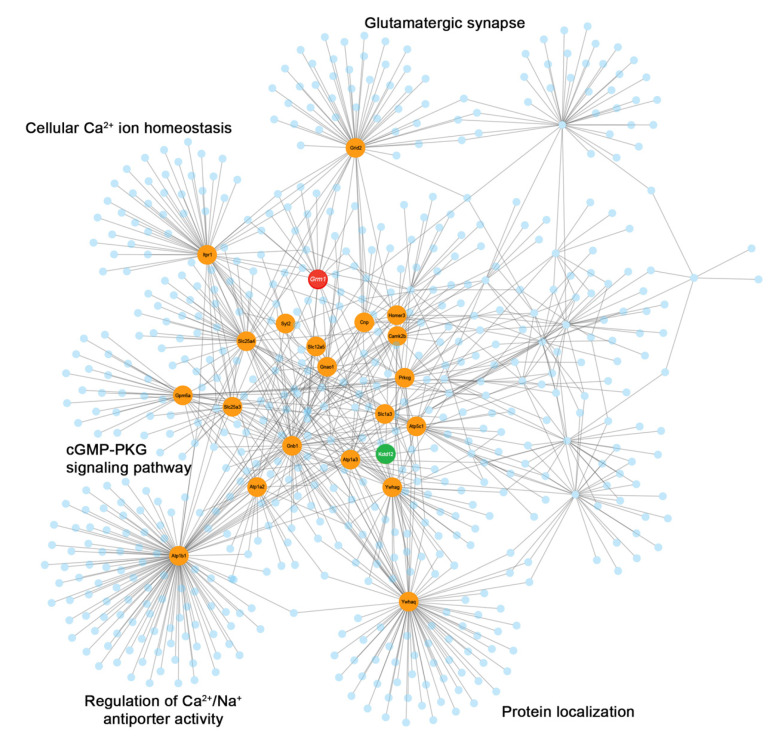
Protein network associated with mGlu_1_ in mouse cerebellar synapses. The nodes represented by high-confidence mGlu_1_ putative and established interactors are shown in orange, mGlu_1_ is shown in red, and KCTD12 in green. A few GO-extracted functionalities are given at the side of protein communities having a high confidence mGlu_1_ interactor as the starting protein.

**Figure 3 cells-12-01325-f003:**
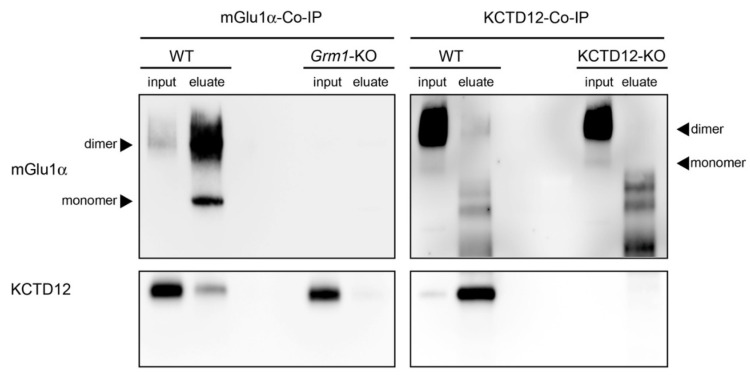
mGlu_1_α and KCTD12 co-immunoprecipitants from the cerebellum. Left: mGlu_1_α was immunoprecipitated from the cerebellum of WT and *Grm1*-KO mice using the anti-mGlu_1_α antibody raised in guinea pigs and immunoblotted using anti-mGlu_1_α (**top**) and anti-KCTD12 (**bottom**) antibodies raised in rabbit. Lanes: WT-input; WT-eluate; KO-input; KO-eluate. Right: KCTD12 was immunoprecipitated from the cerebellum of WT and KCTD12-KO mice using the anti-KCTD12 antibody raised in guinea pigs and immunoblotted using the anti-mGlu_1_α raised (**top**) and anti KCTD12 (**bottom**) antibodies raised in rabbits. Lanes: WT-input; WT-eluate; KO-input; KO-eluate. Non-specific bands were detected between 90 and 70 kDa with the anti-mGlu_1_α antibody in eluates of KCTD12-Co-IPs.

**Figure 4 cells-12-01325-f004:**
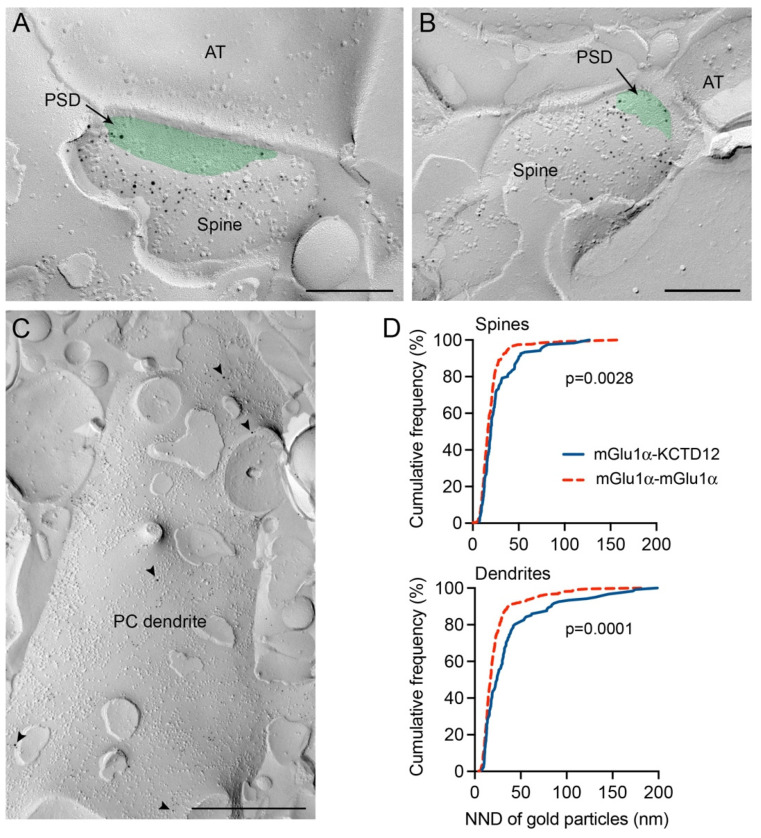
FRIL shows co-distribution of mGlu_1_α and KCTD12 in the same nano-domain at dendritic spines and shafts of cerebellar Purkinje cells. (**A**,**B**) Electron micrographs showing co-localization of mGlu_1_α (5 nm) and KCTD12 (10 nm) in spines. A pseudocolor (green) has been used to simplify the identification of the postsynaptic density (PSD). (**C**) A Purkinje cell dendrite shows KCTD12 labeling at extra-synaptic sites (arrowhead). (**D**) Cumulative distributions of the nearest neighbor distance (NND) of gold particles detecting mGlu_1_α and KCTD12 in Purkinje cell spines and dendritic shafts. Abbreviations: AT, axon terminal; PC, Purkinje cell. Scale bars: (**A**,**B**) 200 nm and (**C**) 500 nm. Data were analyzed by means of the two-sample Kolmogorov-Smirnov Test. *p* values < 0.05 were considered significant.

**Figure 5 cells-12-01325-f005:**
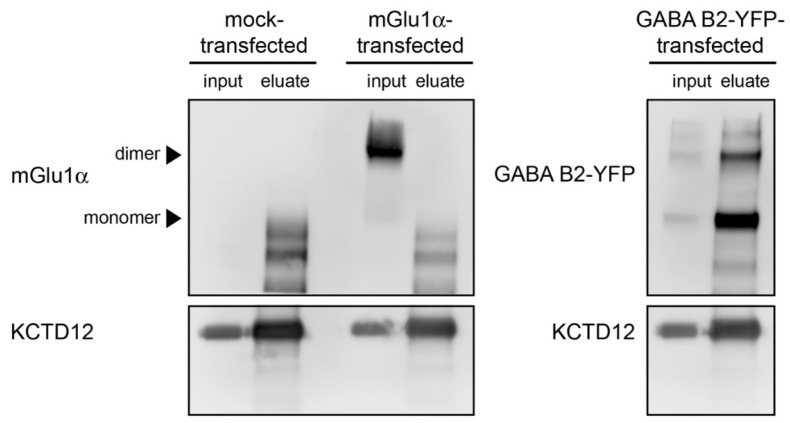
mGlu_1_α and KCTD12 do not directly interact *in vitro*. KCTD12 was immunoprecipitated from KCTD12 cells that were not transfected, transfected with mGlu_1_α, or transfected with GABA_B2_-YFP plasmids. The anti-KCTD12 antibody raised in guinea pigs was used for immunoprecipitation and the anti-mGlu_1_α (**top left**), anti-GABAB2 (**top right**), and anti-KCTD12 (**bottom**) antibodies raised in rabbit were used for immunoblots. Lanes are from whole cell lysates of mock-transfected-input; mock-transfected-eluate; mGlu_1_α transfected-input; mGlu_1_α transfected-eluate; GABA_B2_-YFP transfected-input; and GABA_B2_-YFP transfected-eluate. Non-specific bands were detected between 90 and 70 kDa with the anti-mGlu_1_α antibody in eluates.

**Figure 6 cells-12-01325-f006:**
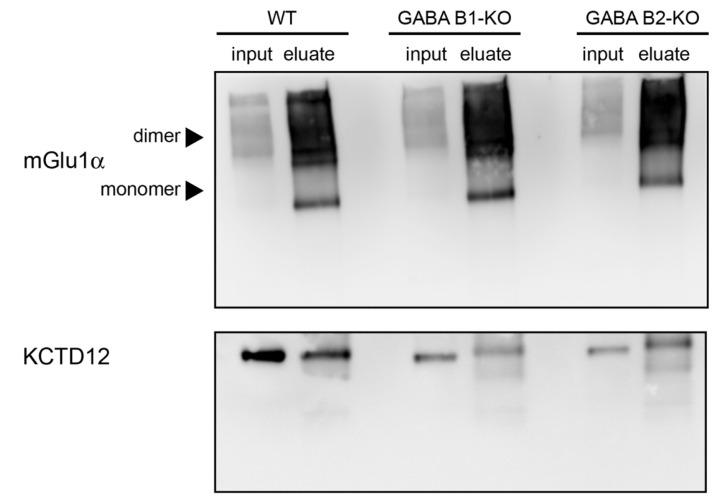
The interaction between the mGlu_1_α and KCTD12 is not mediated by GABA_B_ receptors. mGlu_1_α was immunoprecipitated from WT, GABA_B1_-KO, and GABA_B2_-KO cerebellar lysates. The anti-mGlu_1_α antibody raised in guinea pigs was used for immunoprecipitation. The anti-mGlu_1_α (**top**) and anti-KCTD12 (**bottom**) antibodies raised in rabbits were used for immunoblots. Lanes: WT-input; WT-eluate; GABA_B1_-KO-input; GABA_B1_-KO-eluate; GABA_B2_-KO-input; GABA_B2_-KO-eluate.

**Table 1 cells-12-01325-t001:** Identification of mGlu_1_α by LC–MS/MS in six immunoprecipitation experiments.

Coverage (%)	Unique Peptides ^1^ (Number)	Peptides (Number)
35.11	11	33
31.61	7	29
43.20	10	34
49.54	11	39
46.79	11	37
31.61	8	27

^1^ Peptides not shared with mGlu_1_β.

**Table 2 cells-12-01325-t002:** mGlu_1_ interacting proteins identified in all six immunoprecipitation experiments.

Protein ID	Protein Name	Gene ID	Unique Peptides (Number)	NCBI ID
P97772	metabotropic glutamate receptor 1	*Grm*1	48	NP_058672.1
P18872	guanine nucleotide-binding protein G(o) subunit α	Gnao1	11	NP_034438.1
P62874	guanine nucleotide-binding protein G(I)/G(S)/G(T) subunit β-1	Gnb1	6	NP_001153488.1
Q8BW86	ρ guanine nucleotide exchange factor 33	Arhgef33	14	NP_001138924.1
Q6PIC6	sodium/potassium-transporting ATPase subunit α -3	Atp1a3	26	NP_001361556.1
Q6PIE5	sodium/potassium-transporting ATPase subunit α-2	Atp1a2	14	NP_848492.1
P14094	sodium/potassium-transporting ATPase subunit β-1	Atp1b1	7	NP_033851.1
Q91V14	solute carrier family 12 member 5	Slc12a5	29	NP_001342409.1
P56564	excitatory amino acid transporter 1	Slc1a3	6	NP_683740.1
O35544	excitatory amino acid transporter 4	Slc1a6	7	NP_033226.1
Q61625	glutamate receptor ionotropic, δ-2	Grid2	20	NP_032193.1
Q99JP6	homer protein homolog 3	Homer3	14	NP_001139625.1
P16330	2′,3′-cyclic-nucleotide 3′-phosphodiesterase	Cnp	10	NP_001139790.1
Q91VR2	ATP synthase subunit γ, mitochondrial	Atp5c1	6	NP_065640.2
Q8VEM8	phosphate carrier protein, mitochondrial	Slc25a3	5	NP_598429.1
P48962	ADP/ATP translocase 1	Slc25a4	3	NP_031476.3
P61982	14-3-3 protein γ	Ywhag	7	NP_061359.2
P68254	14-3-3 protein θ	Ywhaq	8	NP_035869.1
P11881	inositol 1,4,5-trisphosphate receptor type 1	Itpr1	62	NP_034715.3
Q7TNC9	inositol polyphosphate-5-phosphatase A	Inpp5a	10	NP_898967.2
P63318	protein kinase C γ type	Prkcg	16	NP_035232.1
P28652	calcium/calmodulin-dependent protein kinase type II subunit β	Camk2b	4	NP_001167524.1
P35802	neuronal membrane glycoprotein M6-a	Gpm6a	2	NP_705809.1
P46097	synaptotagmin-2	Syt2	4	NP_001342655.1
Q6WVG3	BTB/POZ domain-containing protein KCTD12	Kctd12	9	NP_808383.3

## Data Availability

Proteomic data are available via ProteomeXchange with the identifier PXD040886.

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
