# Peer review of "Protein Networks Associated with Native Metabotropic Glutamate 1 Receptors (mGlu1) in the Mouse Cerebellum"

_cells, 2023, doi:10.3390/cells12091325_

Round 1
Reviewer 1 Report
The mouse cerebellum mGlu1 interactome study by Mansouri et al identified KCTD12 as consistent indirect interactor, among 25 proteins that were invariably coimmunoprecipitated with mGlu1 in 6 experiments with two different antibodies. Control experiments using Grm1-KO and KCTD12-KO cerebellum, as well as validation by electron microscopy and HEK293 transfection, supported this conclusion. The project was conducted by experienced researchers with maximal diligence, and is presented excellently. I have no major criticism, only very minor suggestions:
(1) In paragraph 2.10: Which software was used to visualize the network?
(2) Abbreviations should be defined in the flow text: In paragraph 3.5. FRIL, in Figure panel 4D NND. A typo in Figure 4D X-axis: particles instead of paticles.
(3) Usually, the authors wrote in correct manner number with units separated by a space, and numbers with a percent sign without intervening space. However, sometimes they had lapses that merit correction:
Methods paragraph 2.4: 6 oC (1:3,000).
Paragraph 2.5: typographical error: LTQ Orbitrap XL. 1% false discovery rate.
Paragraph 2.6: 37 oC
Paragraph 2.7: 6 oC
Paragraph 2.9: 72 hr at 6 oC, .... 48 hr at 6 oC
Reviewer 2 Report
The original article entitled “Protein Networks Associated with Native Metabotropic Glutamate 1 Receptors (mGlu1) in the Mouse Cerebellum” by Mansouri et al. describes the multiprotein network that associates with mGluR1 receptor in cerebellar tissue. The main goal of the manuscript is to decipher proteins that interact with mGluR1 directly and indirectly forming multiprotein complexes. To this end Authors use proteomic mass spectrometry followed by coimmunoprecipitation, electron microscopy and western blotting to validate direct interaction between mGluR1 and novel putative partner KCTD12. Receptor mGluR1 is crucial for excitatory synaptic transmission, excitatory synaptic plasticity and heterosynaptic GABAergic plasticity. It is worth noting that the manuscript extends our knowledge on mGluR1 into the analysis of its interacting partners
The manuscript is informative and clearly written. Presented discoveries would be suitable for researchers of synaptic transmission and synaptic plasticity.
I have only one minor comment as listed below:
1. Figure 2 is in my opinion unclear and uninformative. Authors should consider changes in this figure to incorporate at least protein names (instead of IDs) and increase the clear information content for regular reader.
